# Europeanization as Pragmatic Politics: Italy's Civil Society Actors Operating in the Face of Right-Wing Populism

**Fazila Mat** [1,*], **Luisa Chiodi** [2] and **Oliver Schmidtke** [3]

1. Department of Political Science, University of Victoria, 3800 Finnerty Road, Victoria, BC V8P 5C2, Canada
2. Osservatorio Balcani Caucaso Transeuropa—CCI, Via Tartarotti, 7, 38068 Rovereto, Italy; chiodi@balcanicaucaso.org
3. Centre for Global Studies, Department of Political Science, University of Victoria, Victoria, BC V8S 1L4, Canada; ofs@uvic.ca
* Correspondence: fmat@uvic.ca

**Abstract:** This article examines how and under what conditions Italy's civil society organizations (CSOs) have resorted to transnational activism and to what extent these efforts translate into impactful political advocacy. The analysis focuses on the action strategies of these civil society actors that have come under considerable pressure through the resurgence of populist–nationalist actors in the domestic arena. Developing an actor-centred perspective from below, this article draws on a series of 27 interviews conducted with these organizations' representatives working primarily on issues related to migration and refugees in Italy. The empirical study examines some key initiatives that see domestic CSOs as protagonists in the transnational realm and explicates their motivations, approaches, and experiences. Conceptually, the article distinguishes between the vertical and horizontal Europeanization of CSOs. While there are notable opportunities for CSOs to engage in Brussels-centred governance and policy making, the effectiveness of horizontal Europeanization in the form of cross-border networking is—at first sight paradoxically—limited by the EU's system of multi-level governance. The central argument about Europeanizing civil society activism is that these processes are primarily driven by a pragmatic pursuit of solutions to concrete political challenges that could not be properly addressed in an increasingly hostile domestic environment.

**Keywords:** civil society; Italy; Europeanization; populism; transnationalization; migration; Europe

## 1. Introduction

The rise and gradual entrenchment of right-wing populist and nationalist parties have transformed Europe's political landscape dramatically. This fundamental change concerns central features of competitive party politics (Albertazzi and Vampa 2021; Schmidtke 2023) as well as the environment in which civil society organizations operate and add to the vibrancy of liberal democracy (Ruzza and Sánchez-Salgado 2021). The resurgence of populism has led to a shrinking space for opposing political voices, claims for the supremacy of state authority, and declining opportunities to promote the rights of marginalized groups in society. In a nutshell, right-wing populism's project to promote the unified and homogenous voice of the 'people' tightens the political space for dissent (Urbinati 2019).

Italy is a case in point. Over the past two decades, the country has witnessed the resurgence of right-wing populism, a prevalence of state-sponsored nationalism and the stigmatization of migrants (Basile and Borri 2022). It is against this background that this article examines civil society organizations (CSOs) advocating primarily for fundamental civic and human rights. Based on a series of 27 in-depth interviews with representatives of CSOs, we reconstruct the growing political constraints that they face in carrying out their political advocacy and analyse what political options they pursue with respect to exploring opportunities at the transnational, European level. At the core of our investigation is the

question if, why, and in what form civil society organizations have trans-nationalized their approach in order to regain some of their waning influence in the domestic political arena.

Starting in the early 2000s, the concept of transnationalism has gained traction to describe how politics cannot simply be understood in the confines of national communities. In particular, in the European Union, the dynamics of cross-border interactions have reshaped the political landscape and opened up opportunities in a multi-layered governance structure in which the national arena is a critical, albeit not exclusively relevant, political space. To consider the political implications of transnationalizing processes with a focus on CSOs may seem paradoxical at first sight: these organizations are very much place-based and regularly rooted in specific local realities. Still, in the Italian context, these CSOs have demonstrated a deepened commitment to exploring political opportunities at the international, European level (Chiodi 2021; Crepaz 2022; della Porta 2022; for an overview see: Mattoni and Rone 2022).

The academic literature analyses transnationalization primarily in light of the ongoing globalization process that make activists' cross-border interactions more frequent and more intense, fuelled by cheaper and faster travel and the advent of the internet. This process has not only provided opportunities for instantaneous communication but also reduced the role of civil society gatekeepers and augmented the repertoires of action available to activists in virtually all parts of the globe (Ford 2022). Yet, this article intends to make an argument that goes beyond the general ideas about the effects of globalization and cross-border exchange. It claims that civil society activism at the transnational level is primarily driven by pragmatic responses to political challenges that cannot be adequately addressed in the domestic arena alone. Reconstructed mainly with the help of semi-structured interviews with key stakeholders from Italy's civil society, we examine what forms of transnationalization different types of CSOs have explored and what the specific effects of these efforts have been on the advocacy they seek, with particular regard to migrants and asylum seekers.

This article will proceed in three steps. First, we discuss the conceptual framing of transnationalization and its meaning for civil society actors in the European context. This section will allow us to highlight key assumptions about Europe's evolving political landscape operating at different levels of governance. Second, the article briefly sketches the political opportunities for Italy's civil society organizations in an environment that is increasingly shaped by right-wing populist, anti-immigrant parties. Third, in the main part we analyse different attempts at transnationalizing efforts through vertical and horizontal coordination across national boundaries. The concluding section then seeks to assess the nature, depth, and effects of transnationalizing CSOs' advocacy efforts in Europe.

## 2. Conceptualizing Transnationalization in the European Context

Transnationalism, understood as a 'people-led process' or action 'from below' (Smith and Guarnizo 1998), involves civil society and individuals moving across borders as well as their formal and informal activities, which often entails 'goal-oriented initiatives' (Portes et al. 1999; Kearney 1995; Peck 2020). These projects include "working to promote or oppose some form of political change internationally or across borders" (Ford 2022, p. 170). The term 'transnationalism', however, is a contested one since its meaning is highly dependent on its specific context. Scholars have noted different paths of scale-shifts (downward and upward; Tarrow 2005), different forms of diffusion (thin and thick; della Porta and Mattoni 2014) as well as different types of connections (networked versus aggregative; Vicari 2016). Similarly, they have also pointed at 'different forms' of transnationalization, involving on the one hand "more hierarchically oriented NGOs" and "horizontal social movement networks" on the other (della Porta 2018, p. 351).

The literature on the transnationalization and Europeanization of CSOs has adopted a strong and central focus on protest and other forms of contentious action as an indicator of the transnationalization of social movements (della Porta and Caiani 2009; della Porta and Tarrow 2005; Imig and Tarrow 2000; Koopmans and Statham 2010). The social

movements literature has typically placed civil society actors and social movements in different domains, with the latter considered as embodying contentious and the former non-contentious forms of engagement.

More recently, della Porta and Steinhilper (2021) have argued that "dynamics of hybridisation" between the two forms of collective action have become more pronounced in Europe following the series of crises that have affected the continent since the financial upheaval in 2008. In particular, they noted that the 'refugee crisis' in 2015 and the rise of populist and nationalist parties or movements have restricted both migration and the space of civil society, transforming them into "two sides of the same coin" (della Porta and Steinhilper 2021, p. 177). The same crises over the last decade have also affected progressive civil society organizations' transnationalization in the European context (della Porta 2020). Studies have argued that civil society activism increased Europeanization of the early 2000s coinciding with a growing retrenchment at the national and local levels (della Porta 2020; Kaldor et al. 2015) significantly driven by the rise of Euro-sceptical forces in many European countries (Pirro et al. 2018).

There has also been a notable shift in the scholarly literature from a focus on Brussels-based CSOs towards domestic civil society mobilization in Europe (Sánchez-Salgado and Demidov 2018; Wunsch 2018; Novak and Lajh 2018; Odasso 2018; Buzogány 2018). Indeed, we posit that the domestic context plays a fundamental role for the transnational mobilization of CSOs. These organizations target multiple entities at different scales either simultaneously or over time, but their existence and work is inevitably shaped by the conditions—physical, legal and social—imposed by particular states (Tarrow 2005; Zajak 2017). At the same time, the Brussels-based NGOs face difficulties when seeking to trickle down their action to the domestic level and remain detached from national political dynamics (Quittkat 2011). Moreover, as stressed by Ruzza and Sánchez-Salgado (2021), the populist turn has rendered much of the literature on the role of CSOs in Europe obsolete. CSOs have been under attack at the nation–state level, particularly in countries with a strong populist representation in government, such as Italy.

*2.1. CSOs' Europeanization in Europe's Multilayered Governance Structure*

The concept of Europeanization has long been dominated by the 'downloading' approach, which is understood as a 'top-down' pressure wherein European-level policies affect national as well as regional or local policies (Buller and Gamble 2002; Bursens and Deforche 2008; Knill and Lehmkuhl 1999; Graziano and Vink 2008). According to Börzel (1999, p. 574), Europeanization is "a process by which domestic policy areas become increasingly subject to European policy-making".

At the core of the Europeanization literature is the debate on how best to conceptualize the link and interaction between national and European levels of governance. Scholars who attribute priority to the European level argue that the participation of CSOs follows a transmission belt model, capturing that EU-based CSOs could be transmitters of EU information to domestic audiences, while acting as a collector of knowledge and information from the domestic level and bringing it into the EU debate (see Buonanno and Nugent 2020).

There are multiple studies depicting how the EU both has indirectly and directly mobilized and utilized civil society associations in various policy fields where it was deemed productive to promote such an agenda (Bouwen 2009; Coen and Richardson 2009; Sánchez-Salgado 2007). The EU's forms of financial, technical, and ideational support have encouraged scholars to talk of "participatory engineering" (Zittel and Fuchs 2007) in order to capture this "active and imaginative" bureaucratic activism (Kohler-Koch and Finke 2007, p. 205). Studies have also explored and pointed out the selective relationship between the EU and certain categories of groups privileged over others (Armstrong 2002; Greenwood and Halpin 2007).

According to an EU-focused framework, the Europeanization of CSOs is mainly driven by Brussels-centred policy priorities. These approaches imply that government actors will engage CSOs either because they are pressured to do so by the EU, or because CSOs

possess specific political capital, knowledge or expertise that make them useful partners for policy reforms (Ruzza and Sánchez-Salgado 2021). In this sense, studies adopting the 'downloading' approach have been criticized for disregarding the role of political actors and downplayed their agency. Even when these actors are recognised as 'norm entrepreneurs' that partake in a learning process and contribute to the diffusion of ideas and norms in the domestic context (Caporaso et al. 2001; Börzel and Risse 2003), they are still considered as 'mediators' "trapped between the misfit pressure and institutional change" (Jacquot and Woll 2003, p. 3).

The 'downloading' approach to Europeanization has been challenged by studies that instead highlight a 'bottom-up' approach, centred on the domestic level (e.g., conditions, structures and actors; Radaelli 2003) where civil society is considered a primary actor (Trenz 2011). For instance, della Porta and Caiani describe 'bottom-up' Europeanization as a process of "Europeanization of and by civil society" (della Porta and Caiani 2009, p. 25) and see it as a tool to counter the EU's "democratic deficit" (Follesdal and Hix 2006). Research in bottom-up Europeanization focuses on understanding the use of the EU by civil society actors in framing their concerns and changing unfavourable domestic opportunity structures through capacity building (Trenz 2011, p. 173).

We consider our study to be in the tradition of studies based on such a bottom-up understanding of Europeanization that takes the domestic level as its starting point. We build on the 'usage of Europe' concept focusing our attention on how the EU is instrumentalized by domestic actors (Jacquot and Woll 2003; Graziano et al. 2011). In this sense, we interpret the EU primarily "as a selective amplifier rather than as the key driver of change" (Sánchez-Salgado 2014, p. 19).

The EU provides a context, cognitive and normative frames as well as opportunities for the socialization of domestic actors, who, in turn, produce an exchange of ideas, practices, and policies. However, while the EU offers material and non-material resources to domestic actors, much depends on the use of these by the actors in question. Radaelli and Exadaktylos's (2010, p. 193) definition of Europeanization as "an interactive process, rather than a simple process of unidirectional reaction to Europe" proves particularly productive for our research questions. It covers both the notion of Europeanization as 'domestic impact of Europe' (or pressure) and Europeanization as creative usages of Europe in the domestic arena. In addition, in the empirical analysis we consider 'horizontal Europeanization' as a way for CSOs to engage in forms of mutual learning, cooperation and support across the borders of EU member states.

### 2.2. Empirical Data and Methodological Approaches

The primary source of this research is 27 semi-structured, in-depth interviews with representatives of civil society organizations, activists, and lawyers based in Italy. The interviews were conducted in two phases, between July–August 2021 and July–November 2022. We completed sixteen interviews online during the first phase, in a period still characterized by mobility restrictions imposed by the COVID-19 pandemic. The identities of the interviewees have been kept anonymous to enable them to speak more freely. We complemented the content analysis of the interviews with documentary evidence from campaigns in the form of press releases, reports, media coverage, and judicial decisions.

In our interviews, we addressed broad questions, probing and adding follow-up questions on the spot, according to the nature of the individual response. Our goal was to observe the subjective meanings of phenomena attributed by the people experiencing them. We first asked the interviewees to express their views on the reasons for the attacks against and criminalization of CSOs by populist and anti-immigration politicians following the 2015 'refugee crisis'. In a second step, we prompted them to elaborate on the effects of these attacks on their work. We then asked our interlocutors to talk about the political context in Italy and to tell us how and why they think they are or are not able to have a voice in the national political agenda[1].

Since we intended to probe what kind of CSOs are engaged in Europeanization efforts, we have interviewed representatives of civil society organizations of different sizes and capacities. Firstly, we talked to members of highly organized associations, some of which have been active for decades and act as regular interlocutors of state institutions on issues central to their mission. Secondly, we interviewed representatives of organizations that emerged only a few years ago with the specific purpose of countering mounting discriminatory policies heralded by national and local governments.

A few of the selected organizations are the Italian branches of wider European or international organizations, while others represent networks of national and local associations. The third and fourth groups, respectively, comprise small grassroots initiatives and medium-sized organizations, some of which are formed by professionals who voluntarily dedicate their time to foster the cause of their association. In terms of their specific focus, 12 of the organizations contacted work exclusively on issues related to migrants, including providing legal support and advancing their position in society and/or by raising public awareness on these issues. Three organizations strive to enhance LGBTIQ+ and women's rights, while the remaining twelve are mostly human rights and anti-racist organizations, with many of them attempting to foster economic and social justice, as well as citizenship rights and grassroots participation.

The purpose behind this selection was to explore how and to what extent these organizations joined their forces and deployed common strategies for furthering fundamental rights at the transnational level in the face of Italy's right-wing political turn. Our working hypothesis was that the size, competences, and capacities of these CSOs affect the extent and form in which they pursue Europeanization. When selecting the cases, we did not focus on the dichotomy between contentious and conventional political tactics often ascribed to social movement organizations and non-governmental organizations, respectively (della Porta 2018). Instead, we followed the literature that notes the hybrid character of transnational political action, where institutionalized and non-institutionalized forms of organization and action are adopted both by grassroots and more structured organizations (Roy 2011; Alvarez 1999).

## 3. Delegitimizing Italy's CSOs in the Wake of the Populist–Nationalist Resurgence

Civil society actors have traditionally been a bedrock of Italy's post-war democracy. Initially closely tied to the country's main parties and polarized political camps during the Cold War, CSOs have taken on a new role in a political system characterized by declining trust in mainstream parties. The volatility in Italy's party politics has created a paradoxical situation for civil society in promoting democratic participation and citizens' engagement. On the one hand, Italian civil society has gained autonomy from party politics and emerged as an independent voice in public and policy debates over the past two decades. On the other hand, this process of empowering CSOs has partly been driven by a weakening of political parties as the main agents of liberal democracy, and new challenges to the democratic process as a whole. Over time, civil society found itself without an institutional counterpart capable of transforming its requests into political decisions and policy processes: "We find it increasingly difficult to identify allies in the political field" (Int. 9)[2].

The rise of anti-immigrant and populist parties has further transformed the legal and social conditions under which the CSOs operate. One critical dimension of this transformation has been the de-legitimization of CSOs themselves. In particular, following the so-called 'refugee crisis' in 2015–2016, Italian political elites started to slam non-governmental organizations (NGOs) engaged in rescuing migrants at sea as part of Search and Rescue operations in the central Mediterranean, accusing them of being complicit in the mass arrivals of migrants (Cusumano and Villa 2021; Krzyżanowski et al. 2018).

In 2017, the Minister of Interior of the Democratic Party, Marco Minniti (a member of the centre-left government led by Paolo Gentiloni), introduced the Code of Conduct for NGOs (Ministero dell'Interno n.d.), which was voluntarily signed by two search and rescue

organizations, while the others refused to accept it, facing exclusion from official rescue operations in the Mediterranean. Two rules were mostly criticized. The first required that NGOs avoided sailing into Libyan waters (unless lives were at risk) and interfering with the rescue operations of the Libyan coastguard,—which were often accused of violating migrants' rights. A second rule that was equally opposed by the organizations was the potential presence of armed police on board the NGOs' vessels (Sarzanini 2017).

The Code of Conduct was followed by the restrictive policies put in place between June 2018 and August 2019 by the subsequent government, formed by an alliance between Matteo Salvini's far-right Lega and the anti-establishment Five Star Movement (M5s)[3]. The new Italian government engaged in "a real tug-of-war, both with regard to the NGOs, accused of directing migratory flows towards Italy, and with regard to other European States, guilty of leaving Italy alone in disembarking and receiving migrants" (Cancellaro 2020, p. 201).

As noted by Pusterla (2021), initially, the measures adopted against NGOs were meant to "dissuade" them from helping migrants. In contrast, the government under the Lega and the Five Star Movement tightened the national legislation, turning the "criminalization of solidarity from a political to a legal dimension" (p. 79). However, it is worth mentioning that the first criminal investigations against CSOs that provided help and relief to migrants in transit date back to 2016, during Gentiloni's centre-left government.

> "In Italy the thing that worried us a lot was not only the specific issue, i.e., the criminalization of those who provided rescue at sea, but then by extension the delegitimization of anyone who did not fall into the private or public category, therefore NGOs, social movements, anyone who did not recognize themselves in being a private or a public enterprise." (Int. 4)

The media also played a key role in shaping public opinion by making politicians framing immigration as a source of insecurity the dominant perspective framing the issue. While Italian state television RAI featured a distinct "pro-government bias" (D'Arma 2015, p. 124), national channels owned by the media tycoon and former prime minister Silvio Berlusconi, provided another platform for "establishing Salvini in the position of main protagonist on migration issues" (Maneri et al. 2023, p. 82; see also Amnesty 2020). However, as one interviewee noted (Int. 8), even the more progressive press contributed to this hostile atmosphere that eroded the Italian citizens' trust towards CSOs.

As a consequence, the positive connotation of the world of association that Italians have traditionally had has also been negatively impacted: *"I believe that the campaign made by Minniti, starting from the NGO code, has created a distance between a good part of public opinion and the associations, and I don't know if we will ever bridge the gap"* (Int. 5). As another interviewee vividly put it: *"We had a baptism in the left, with a continuation in the right and the public opinion was eventually convinced on both sides"* (Int. 8).

Within this changing political context, CSOs often found themselves directly targeted *"by political leaders of some very clear political factions"* and not supported by state institutions which *"did not sufficiently oppose the criminalization of defenders' of migrants' rights"* (Int. 4). The feeling of being abandoned and politically marginalized even by political interlocutors with whom CSOs had traditionally interacted is exemplified in the words of another interviewee, who mentions the difficulty of hosting political representatives in public events concerning migrants:

> "In recent times, to be honest, there has been a certain reluctance that has been somewhat telling in itself, a sort of hesitancy from politicians to accept our invitation. We do keep inviting them to take part, but in recent years there has sometimes been a sort of rejection." (Int. 14)

The de-legitimation of the CSOs in the national political setting became manifest in other ways as well. For example, they have been overwhelmed with endless bureaucratic requests or invited to take part in the political decision-making, only to be ignored or have little attention paid to their proposals (Int. 4). In this sense, there is a general perception

that *"there is lack of translation"* of CSOs' efforts because the domestic political setting *"is not able, or does not want to or is not interested"* (Int. 1).

In sum, the 2015/2016 'refugee crisis' and its political ramifications contributed to a political environment in which human rights oriented CSOs felt increasingly isolated and deprived of some of the venues through which they used to channel their advocacy. The broader political implications of the criminalization campaign has been that the whole non-governmental sector ended up being negatively portrayed and stigmatized. With the narrowing of political opportunities domestically, Italian CSOs needed to find new avenues to regain their political relevance and influence.

## 4. Italy's CSOs' Vertical and Horizontal Transnationalization in a European Space

As a reaction to the closing of political opportunities at the national level in the context of discriminatory policies, fundamental rights violations, and the criminalization of solidarity (Mainwaring and DeBono 2021), CSOs were driven to look for transnational solutions related to the EU's system of multilevel governance. In considering transnationalization efforts by CSOs, it is worth underlining that 'Europe' is welcomed by them as an integral part of exercising citizens' democratic rights and practicing solidarity. Our interviews provide clear evidence that the European governance level has become more attractive as right-wing populist parties and their nationalistic agendas have gained support:

> "Clearly, considering the right-wing occurrences [in Europe] [from Sweden, to Italy and to Hungary and France] we are faced with a right that is based on very strong identities. I believe there will be a temptation to backtrack on a whole series of issues related to rights, which will start above all from narrative and then possibly transform into regulatory measures. And if this were to happen, Europe becomes useful". (24a)

Exploring legal and political opportunities at the European level has become an increasingly important component of CSOs' advocacy and political campaigns across the continent. The notion that 'Europe becomes useful' is a critical theme that runs through many interviews. The reference to the European level of governance is—to a large degree—instrumental and prompted by the experiences in the domestic political arena. The tools that domestic actors try to exploit range from legal and financial means to cognitive, normative political and institutional resources (Woll and Jacquot 2010; Koopmans 1999). However, domestic actors are not just passive transmitters or receivers of ideas and policies from one political level to another. Rather, they try to use EU resources to pursue their political agenda and might draw on the EU when it is useful to them and their agenda, while in other instances they might not refer to the EU at all (Graziano et al. 2011, pp. 13–14). At the same time, Italy's CSOs have few avenues for financing their activities (Maggini and Federico 2021) and clearly the availability of resources—funds, personnel, and skills—has a profound effect on their capacity to use the instruments offered by European polity.

By looking into the strategies used by CSOs in trying to achieve policy change in the domestic arena, this article focuses on two types of mobilization (vertical and horizontal) in connection to the 'top-down' and 'bottom-up' understandings of Europeanization. In this context, the term 'vertical transnational mobilization' is not intended as a "scale shift towards the EU", achieved by a CSO through its membership in a European or international umbrella organization (Tarrow 2005; Lahusen et al. 2021). We use the term to refer to the links established by CSOs with EU institutions and actors inside their country or at the EU level, taking place in two distinct forms. First, vertical mobilization captures the role of civil society organizations in applying bottom-up "adaptational pressures" to their domestic context (Caporaso et al. 2001). This type of mobilization "reproduces top-down EU pressures from the bottom up" (Wunsch 2018, p. 11) by resorting to the precedence that EU legislation has over the national one.

This supremacy of EU law increases the perceived costs for governments of not conforming to EU demands, and it can result in the member state downloading EU policies and regulations and generating policy change at the domestic level in line with the CSOs'

goals. However, vertical mobilization can also take the form of 'uploading', which occurs when civil society organizations actively work to shape EU policies and regulations with the aim of indirectly affecting the domestic ones. The effectiveness of this strategy mostly depends on the tools and resources available to the organizations at the EU level—such as petitions, lobbying and citizen initiatives—and the European institutions' openness to integrating these preferences at the EU level.

The second type of "transnational bridging" (Holzhacker 2009) employed by civil society organizations is horizontal, involving cross-border mobility and networks of non-state actors (Mau and Mewes 2012). In our understanding, this form of transnationalization entails domestic civil societies learning from each other by sharing goals, strategies, and action repertoires (Holzhacker 2009). Yet, it also involves collaboration across borders to create solidarity networks (Lahusen et al. 2021; della Porta 2018) and engage in monitoring activities. In certain cases, the information thus collected could determine policy change at the domestic level through processes of vertical modes.

### 4.1. Vertical Transnationalization: Legal Mobilization for Migrants' Rights

As highlighted by Jacquot and Woll (2003), the discretion and role of political actors in the adaptation or downloading process should not be underestimated. In the Italian case, this role of actors on the ground is critically significant in the notion of "legal mobilization" (Passalacqua 2022, p. 2) as a tool of CSOs to promote their agenda. This strategy uses the judicial doctrines of supremacy to review the compatibility of national law with EU law. In Italy it has become one of the main tactics deployed by a few selected organizations for protecting the rights of migrants. The goal is to frame and direct the implementation of EU legislation in a way to promote effective solutions to concrete problems addressed by civil society actors, using the Court of Justice of the EU and national courts in the process:

> "Moving in a context in which you have a common regulatory framework to which you must refer to, supranational courts to which you can turn and whose jurisprudence has relevance at an internal level is important because it allows you to act from one level to another and bring these two levels, the national and the supranational, into contact in a much, much simpler way." (Int. 23b)

The 2009 Lisbon Treaty and the transformation of the European Charter of Fundamental Rights into a legally binding instrument advanced this process and introduced the Union into the 'politics of human rights'. This process created a new opportunity for Italy's CSOs that has shifted their action repertoire from the domestic government level to the European Union. In particular, the expanding competences of the EU conferred to the Court of Justice of the European Union (CJEU), which extended its jurisdiction to cover human rights-sensitive fields, have offered the rights' defenders an opportunity for a "decentralized enforcement of EU law" (Passalacqua 2022, p. 7)—with several legal battles resulting in victories over the last twenty years. Indeed, the strategic use of the EU law to uphold migrants' rights had first gained traction during the fourth Berlusconi government (2008–2011)[4] and the rise of the far-right Lega party in 2013.

This trend has continued over time in parallel with the nationalist and populist trends dominating the political agenda after 2015–2016. In the field of migrants' rights, a few associations with expertise in EU law have continued to bring strategic litigations before the courts to protect migrants from discrimination and provide access to welfare programs in Italy. Many of these cases revolved around the Single Permit Directive (2011/98/EU), which guarantees that non-EU workers legally residing in an EU country receive equal treatment with nationals of that country regarding social security benefits. The Italian legislation, however, did not implement the directive correctly, limiting the benefits to migrants with long-term permits only, and the infringement procedure launched by the European Commission was also of no avail. This situation was affecting *"45% of the non-EU citizens living in Italy"* (Int. 19), which accounted for slightly over 3,300,000 people in 2021 (Istat 2021). *"Then we necessarily went to the judge"* (Int. 19). So far, all of the cases brought to court by these organizations have had an outcome in favour of the migrants' rights.

In another case in 2018, CSOs brought a regulation introduced by a municipal administration run by the Lega, which was intended to limit or exclude non-EU families from accessing social benefits with subsidized rates, including extracurricular activities for children. In 2020, the Court of Milan accepted the appeal presented by the families and the CSOs, recognizing the discriminatory nature of the regulation under EU and national law and ordering its annulment.

In the words of an interviewee, this is due to the *"Italian legal system's recognition of the discriminatory nature of provisions which introduce different treatments in access to services between Italian, EU and non-EU citizens"* (Int. 16).

As demonstrated by these examples, civil society organizations faced with "the absence of alternatives for political or social action"[5] (Barbera 2012, p. 21) at home used an opportunity offered at the EU level to halt discriminatory practices, reproducing top-down EU pressures from the bottom up. However, while this type of vertical transnationalism has achieved important results in protecting migrants' rights, it is only available to a few CSOs with organizational capacities and resources that could rely on the pro bono work of idealist lawyers. Indeed, the elevated costs of the legal fees and the level of legal expertise that is required for engaging in litigation limit other CSOs' access to the strategy of legal mobilization.

At the same time, a—potentially—powerful legal instrument based on the supremacy of EU law can also backfire politically, as opposing parties have regularly portrayed these legislative processes as an EU imposition or violation of national sovereignty (Pin 2019). CSOs thus run the risk of nurturing the Euroscepticism that populist–nationalist forces have made the trademark of their political identity.

Until now, there have been significant direct effects of EU law as it has been applied by the Italian national courts. For the work of the CSOs in Italy, it is important to note that EU provisions in the anti-discrimination field have offered more advanced protections than the national ones. Still, as stressed by one interviewee: *"I am not sure how long the EU's egalitarian push will last. So far it has had a significant equalizing effect in Italy"* (Int. 19).

Considering the great power delegated to domestic supreme or constitutional courts on how EU law is to be applied nationally, some of our interlocutors mentioned the risks posed by sovereigntist political parties to judicial independence in other member states—*"just like in Poland"* (Int. 21) or *"as pushed by Marine Le Pen in France"* (Int. 19). They suggested that an outcome like this cannot be completely ruled out for Italy—*"right-wing parties could have discretionary power on the judiciary and the Constitution"* (Int. 21). Still, our interviewees tend to exclude this type of outcome, indicating the *"EU path"* as *"the only viable path"* to exit from a similar *"dreadful scenario"* (Int. 21).

*4.2. Vertical Transnationalization: Shaping the EU's Agenda*

The EU's intervention in the field of regulating migration and refugees has recurrently triggered a negative response in member states feeding anti-European sentiments rooted in an exclusionary nationalism. At the same time, the EU itself has recently embarked on promoting a migration pact whose primary rationale is based on the resolute protection of borders rather than the humanitarian concerns of people who have to flee their home country due to prosecution or war. In this sense, the attractiveness of Europeanizing for civil society groups' aspiration is related to the subject matter that they address: "We worked more on Italian territory, we had an Italian focus and therefore we did not work so much at a European level…politically, there wasn't this action (…) but now it's fundamental" (Int. 20).

In the domain of migration and refugees, cross border coordination at the European level almost becomes inevitable in order to protect the rights in particular of asylum seekers. In the following quotation, it becomes apparent how functional needs emerging from a particular political or policy challenge can entice CSOs towards Europeanization:

"(…) then since 2015 2016—because in our case we used to focus more on the internal Italian borders, on what was happening there—we tried to be more

attentive to what was happening in neighbouring countries and therefore to build the network with the French, the network with the Swiss, the network with the Austrians and so on." (Int. 23b)

In this context, "Italy's strategic role in the Mediterranean" (Int. 20) has also triggered a bottom-up mobilization of CSOs to create policy and regulatory alternatives both domestically and in the EU. The majority of CSOs that we have interviewed are aware of the tools and opportunities provided by the EU in this regard—such as the European Citizens Initiative or lobbying Italian Members of the European Parliament (EP) to push legal proposals or presenting petitions at the EP: "we use all the tools that can be used to put pressure also on Parliament, on the Commission" (Int. 7).

CSOs perceive the coordination of political action across national borders and the cooperation with like-minded civil society organizations across the continent as a 'natural' addition to their action repertoire. This strategic orientation is also fostered by "a generational evolution at the European level", which comes with CSOs having an increasing number of representatives "with experience acquired abroad and a more European way of seeing things" (Int. 20).

Many CSOs that we interviewed establish this type of cooperation through the pursuit of common projects with other European CSOs or by being members of EU umbrella organizations. Similarly, many organizations find a way of expressing their positions through EU-led consultations—often in coordination with other national CSOs—while others contribute to reports written for European institutions or agencies. These forms of transnationalization are particularly relevant for smaller organizations, which often lack the know-how as well as the financial and human resources to engage in more demanding and potentially more effective advocacy actions:

> "The European perspective is eternally present because for many years, even with the reports and campaigns we have promoted on the topic of external borders, we have been calling for a European dimension. (. . .) There are obviously specific advocacy actions on which networking is simpler: if you write an appeal, if you write an open letter it is clear that you can easily network, but more structured initiatives are much more complex". (Int. 2)

A key aspiration of the CSOs is to change legislation at the European level through a coordinated approach with allies from other countries. In the field of migration policies, there has been a sustained effort to influence the EU's approach to governing its borders and to bolster the rights of migrants:

> "Now we are focusing above all on the possible reform of the Schengen border code, because it is a reform that (. . .) could impact both external and internal borders. And because it could introduce at a European level the same mechanism that we have today for readmissions from Italy, Slovenia from Greece." (Int. 23b)

Integral to this endeavour is the aspiration of CSOs to generate broad public visibility and awareness for issues through cross-national cooperation and exchange:

> "The challenge was to be able to start from very small interventions, which called into question the approach of European policies, and instead bring the big issues to the attention of Italian and European public opinion. (. . .)." (Int. 22)

The national and European governance levels can be closely related in the actual practice of CSOs, as it has become apparent in the evolution of the national campaign "Ero Straniero" (I Was a Stranger)[6] into the European Citizens' Initiative (ECI) "We are a welcoming Europe". Launched in 2017, the campaign "Ero Straniero" presented a citizen-agenda initiative to Parliament to change immigration policies in Italy, succeeding in collecting over 70,000 signatures. This network of CSOs was then invited to take part in the ECI by the Belgian NGO Migration Policy Group (MPG):

> "Clearly we immediately accepted, partly because we already had the network and the experience of collecting signatures ready (. . .) we were all still very

excited at the idea of being able to do something at European level. (. . .) in short, it seemed to us to be an almost obligatory step, therefore not only concerning national legislation, but also trying to change European legislation." (Int. 22)

The impact of this form of transnationalization is not bound to a particular policy initiative but has the potential of providing domestic CSOs of different sizes and capacities with a heightened sense of political capital, international support, and visibility:

"Ero Straniero has expanded and managed to incorporate many other organizations, including those that do not only deal with immigration. So certainly the first legacy [of the ECI] is fostering our ability to mobilize and also its recognition at the level of civil society, also means being able to talk about denunciation, talk about criminalization which at the time was somewhat the topic that was closest to the heart of many CSOs." (Int. 22)[7]

However, in the eyes of our interviewees, the ECI is also a lost opportunity to create horizontal links between the CSOs across member states: "I can safely say that there was not a European network, that there were individual national networks that carried the initiative forward" (Int. 22). Indeed, while the collection of one million signatures requires a strong capacity for vertical coordination between the Brussels-based promoters and national coalitions, it is not a tool designed to create 'Europe from below': "Everyone referred to Brussels and then there was the sharing of materials, but we were always under the guidance or the intermediation of MPG." (Int. 22).

The ECI addressed "the criminalization of solidarity[8], humanitarian sponsorship and (. . .) the monitoring of human rights violations at the borders" (Int. 22) and "could have made a big difference" (Int. 24b). However, it was not able to collect the number of signatures needed to be considered by the European Commission and despite achieving some "intermediate victories" (Int. 24b) these initiatives did not lead to the expected outcomes, leaving the participants of the campaign disappointed:

"(. . .) Similarly, in December 2018, the European Parliament approved a resolution requesting that the European Commission present a legislative proposal which was aimed at establishing a European Humanitarian Visa. However we know well that it didn't go through and we know how much difficulty we had." (Int. 24b)

The disillusionment about the gap between what can be achieved at the EU level and what has actually been accomplished with respect to issues of (irregular) migration within the wider Dublin system is similarly tangible in the following statement by one of our interviewees:

"We arrived at an acceptable reform in the European Parliament which went beyond the approach based on the country of first arrival. It would have meant a change of approach to solidarity. And therefore to the founding values of the Union. That reform completely failed, because the Commission tore it down. Nothing has changed in the new European pact on asylum and immigration when compared to Dublin. Actually, it has become all externalization. So, the fact that it could not be changed despite having come so close was a big disappointment." (Int. 22)

In these instances, our interlocutors criticize the Commission because it did not submit any legislative proposal establishing a European Humanitarian Visa, focusing its attention instead on the Union Resettlement Framework (Relano Pastor 2020). In other interviews, it is the Council—as being formed by national governments—to emerge as the European institution that is singled out as the main party responsible for the EU's migratory policies (Int. 2; Int. 6). However, among our interviewees, there is a tendency to criticize the *"European political direction"* (Int. 1) as a whole, rather than distinguishing the specific roles and mandates of its different institutions, signalling also a limited knowledge of the EU decision-making processes.

In contrast, in some cases, the presence of Italian representatives at the European institutions emerges as a crucial factor in the facilitation or obstruction of the CSOs' initiatives: *"Obviously this also depends on the Italian presence at the EP or in other European institutions."* (Int. 24b). In general, the European Parliament emerges as better disposed towards CSOs' proposals, although there is also apprehension of *"shifting the balance to the right in the European Parliament's composition"* (Int. 27). For those CSOs with limited lobbying capacities at the EU level, being able to access the EP requires establishing a direct link with selected left- and centre-left wing Italian MEPs (Int. 27; Int. 1; Int. 18) and counting on their "continuous efforts" (Bruni 2018).

The dynamics of Europe's system of multilevel governance also limit the effectiveness of horizontal transnational initiatives. Even after a notable investment of time and resources to set up a transnational initiative like the ECI, what is left are the networks established at the domestic level, as highlighted in the following statement from one of our interviewees:

> "The European network could not resist because it was still aimed at that objective of the agreement. Among the signatories, however, the form, desire and intent to continue working together remained. However, on a national level, not on a European level." (Int. 22)

This limitation pushes CSOs to relate to the EU in a pragmatic fashion, looking at what European institutions offer in terms of practical responses such as legal provisions that are more advanced or effective than the domestic ones or resources that can help them carry out their work. It is no surprise that the vertical transnational action of CSOs, reproducing top-down EU pressures, has been more effective in the European political space. CSOs struggle to have an impact in the intricate EU's multilevel governance system, which leads to disappointment after years of engagement and not many achievements. Our interviewees expressed a clearly articulated ambiguity or, as may be more appropriate, frustration with respect to their ability to affect the 'normative power' of the EU. This frustration is particularly evident among medium-sized organizations as well as CSOs that can rely on a wider network and better resources, which are ultimately capable of carrying out initiatives that can reach the EU institutions.

### 4.3. Horizontal Transnationalization: Mutual Learning and Solidarity across National Borders

At the most basic level, horizontal links allow CSOs to exchange experiences, learn from another and consider joint campaigns and other forms of mutual support. Italy's CSOs have established horizontal transnational links with their European counterparts in the context of the massive influx of refugees into Europe during 2015–2016 and the tendencies of many member states to close their borders to asylum seekers. Those few CSOs providing the so-called 'safe passages' to migrants that wanted to transit from Italy (as their country of first arrival under the Dublin system) to other EU countries well exemplify this type of transnationalization:

> "And already in the autumn of 2015 (...) to create these safe passages we gradually came to know that in every city there was a network, there was an association and there was someone. And it was decided to give even more strength to these safe corridors, to get to know each other better, to talk to each other, to share problematic experiences, but also good practices." (Int. 18)

However, when faced with criminal charges, the same CSOs also started to collaborate on how to present their defences in court: *"We exchanged defenses, and trial reports to understand where to be assertive, and the best way out."* (Int.18)

Establishing transnational solidarity networks appears to have crucial importance for CSOs, both to protect themselves and to counter the anti-immigrant narratives present in different European countries:

> "You cannot hold up a struggle on certain rights issues if you do not have a transnational solidarity network. It is hard to win alone as a single country. The

narratives that dominate the entire continent are the ones that, in the end, produce the norms that are imposed at the national level." (Int. 24a)

Some interviewees, stressing the limited opportunities offered by "the national context", describe partaking in these networks as *"maybe the most vital instrument available"* (Int. 3). Yet, the reference to Europe must be considered with caution: It clearly surfaces in our interviews that these transnational initiatives are not based on Euro-enthusiastic premises. Rather they take place in a period clearly defined by Euroscepticism (della Porta 2020) and a profound frustration with the notion of a 'Fortress Europe' seeking to keep out irregular migrants as the only emerging common denominator among its member states. Horizontal forms of coordination also come with their own limitations. They require time and human resources and when they emerge around a specific project they tend to have a short life, hardly enough for creating durable relationships. Another limitation of horizontal networks at the European scale is the diversity of actors coming together from different national contexts and related challenges in addressing different cultures and domestic priorities. While the interviewees underline the encouraging sense of solidarity among civil society actors, the frustration over coordination challenges in the practice of horizontal Europeanization is palpable:

> "There was great solidarity between all the various associations. Clearly, how should we put it, experiencing the daily emergency, everyone in their own country, the differences from country to country, individual personalities and individual events... clearly it wasn't all smooth sailing. There was not a single large European association to turn to. I would be lying to say that everything went well." (Int. 18)

One critical element in this activism is the collective, transnational task to monitor and report on the violation of migrants' fundamental rights:

> "It is also a direct monitoring activity. Sometimes we are in the field, but much more often we make necessary reference to all the other realities and all the other actors who are in the field every day, from Ventimiglia to Greece, from Trieste to the Italian-Austrian border. For us the network must be understood as a group of entities that act at different levels on the territory or on several territories or at different levels, politically understood as political levels, for us it is fundamental to be able to act, both by carrying out advocacy actions and judicial litigation actions". (Int. 23a)

The type of horizontal activities described in this quote, although rare, sheds light on a form of cooperation that links grassroots initiatives with more structured organizations that have the knowledge and resources to transform solidarity into legal actions, indicating a transformation of the horizontal dynamics into vertical dynamics.

In the work of the Italian CSOs, this on-the-ground advocacy engagement with partners in other member states had the dual purpose of providing guidance for state actors and shaping public opinion and political preferences in the wider public:

> "We try to give evidence to the judge that at the moment in which the Italian authorities made a move towards Slovenia and then towards the Slovenian authorities and towards Croatia, they were in a position to know what would have happened subsequently." (Int. 23a)

While the primary impetus for Europeanizing political campaigns and advocacy efforts is driven by the frustration over severe constraints in the domestic political arena, there is also a tangible sense of 'Europeaness' in the CSOs' engagement. In the interviews, there are recurrent references to a European citizenship model that is considered more desirable or superior to the national one. Indeed, the statements of the Italian CSOs also suggest the formation of European identity built around key political principles:

> "Never since I started dealing with these things have I felt like a truly European citizen. Sometimes I actually forget that I'm Italian." (Int. 18)

When CSOs work horizontally in the struggle against the criminalization of solidarity and for migrants' rights, they indeed experience a sense of European belonging. What we detected in our interviews was the belief and hope in the European governance domain as a guardian of fundamental universal rights and humanitarian principles as they are prominently pronounced in the EU Charter of Fundamental Rights. This aspiration reflects the normative aspirations of the CSOs: *"We couldn't do it if we didn't have this conviction, let's say perhaps even ideological..., on the value of rights."* (Int. 23b). In particular, the horizontal collaboration with like-minded civil society organizations from across the continent nurtured this sense of a shared European identity. At the same time, there is a significant frustration among our interviewees about how robust this 'Europeaness' is in terms of a commitment to fundamental human rights in light of the EU's increasingly aggressive stand towards asylum seekers.

## 5. Conclusions

In Italy, civil society organizations have come under increasing pressure with regard to their advocacy for migrants and refugees. The rise of the populist–nationalist forces and governments over the past decade have severely restricted the political space in which they operate and instilled the need to pursue alternate venues of political advocacy. While predominantly being grassroots and embedded in local socio-political realities, the progressively adverse domestic political environment has made the pursuit of transnational opportunities for these organizations' advocacy increasingly attractive, if not an essential element of their action repertoire.

The analysis of the interviews with representatives of Italy's CSOs shows that, first, Europeanization highlights the closing of political opportunities at the national level where the success of populist politics and of its discriminatory policies, its fundamental rights violations, and its criminalization of solidarity drove CSOs to look for transnational solutions. We identified major political constraints in the domestic political arena as the primary drivers for change and as incentives for pursuing a Europeanization strategy.

Second, our analysis distinguishes the vertical and horizontal directions of the transnationalizing process discussing Europeanization's top-down and bottom-up dynamics. One key element of vertical transnationalization for CSOs is the strategic use of the EU's complex and interwoven legislative arenas and, more specifically, litigation as a vehicle for promoting human rights protection. In this context, we noted how these initiatives are carried out by a selected few CSOs with expert knowledge of the EU legislative framework. We also considered vertical transnationalization in terms of the CSOs' uploading to the European framework by contributing to the EU policy process or legal framework, which occurs when CSOs try to shape EU policies and regulations with the goal of indirectly affecting domestic ones. As for horizontal transnationalization, we explored those initiatives that CSOs undertake in collaboration with interlocutors in other EU member states, overcoming the nation–state barriers and joining forces with domestic partners in other countries. It is worth underlining that opportunities for such transnationalizing opportunities were significantly shaped by the size and capacities of CSOs: While larger, more resourceful CSOs were regularly in a position to engage in vertical forms of transnationalization and to explore avenues for advocacy addressing the EU governance level, smaller organizations primarily took advantage of less taxing and resource dependent horizontal modes of transnationalization.

Thirdly, our analysis points to a seemingly paradoxical finding. In particular related to bottom up transnationalizing dynamics, we highlighted the role of the EU institutional framework in limiting the creation of horizontal networks. For instance, we point to the European Citizens' Initiative as being based on a nation–state logic that does not foresee substantial horizontal transnational interactions. In addition, we consider how Members of the European Parliament respond to their own national electorate; how limited the resources are to stimulate transnational coalition building; and how CSOs have to come to terms with diverse, at times conflicting, requests from different national constituencies.

Essentially, CSOs horizontal transnationalizing initiatives found it extremely difficult to take advantage of Europe's multi-level governance system directly. Rather, these efforts to connect, exchange and collaborate across national borders proved beneficial predominantly, if not solely, for strengthening civil society's capacity and shaping domestic political struggles.

Finally, our research suggests that these transnational initiatives are not based on Euro-enthusiastic premises; rather Italy's civil society organizations initiate transnational actions in a period decidedly shaped by Euroscepticism (della Porta 2020). CSOs tend to pursue their goals and advocacy based on pragmatic strategic considerations that, at their core, are driven by the aspiration to instigate change on the ground. It is one of the surprising and counterintuitive findings of our study that in spite of the frustration with the EU institutional dynamics and critical policy decisions at the European level (militarization of borders, etc.) representatives of CSOs are clearly inclined to endorse a European identity. When CSOs work horizontally to defend migrants' rights and contest the criminalization of solidarity, they indeed experience their European belonging. Through dynamics of horizontal transnationalism, there is an emerging sense of shared fundamental political principles providing inspiration and motivation for an increasingly difficult domestic engagement.

**Author Contributions:** The three authors worked equally on the article and are jointly responsible for the conceptualization, data curation, methodology, analysis, and writing of the text. All authors have read and agreed to the published version of the manuscript.

**Funding:** We would like to acknowledge that this article draws on research supported by the Social Sciences and Humanities Research Council of Canada (grant number 435-2019-0461) and the Erasmus+ Programme of the European Union under the Jean Monnet Network "Transnational Political Contention in Europe" (TraPoCo) (grant number: 620881). The European Community's support for the production of this publication does not constitute an endorsement of the contents, which reflect the views of the authors. The EC cannot be held responsible for any use which may be made of the information contained therein.

**Institutional Review Board Statement:** The study was conducted in accordance with the Human Ethics Standard of the University of Victoria (protocol code 19-0232, approved 6 April 2020).

**Informed Consent Statement:** Informed consent was obtained from all subjects involved in the study.

**Data Availability Statement:** The data presented in this study are available on request from the corresponding author.

**Conflicts of Interest:** The authors declare no conflict of interest.

## Appendix A. Main Themes and Text-Based Categories[9]

**A. Political constraints at the national level**

1. Left-right continuum of policies (44%)
2. Rise of anti-immigrant and populist parties (44%)
3. Expansion of negative framing from SAR NGOs to CSOs (44%)
4. Absence of institutional counterpart (41%)

**B. Drivers of transnationalization**

1. A more developed and binding supranational legal framework at the EU level (37%)
2. Migration as a European issue (33%)
3. Aim to change EU's approach to the EU's external border management (33%)
4. Lack of proper implementation of EU's directives at the national level (11%)

**C. Types of transnational actions at the European level**

1. Interacting with EU institutions (reports; consultations; national MeP members) (59%)

2. Networking with other CSOs in Europe (52%)
3. Advocacy (41%)
4. Legal mobilization/strategic litigation (29%)
5. Solidarity activities (15%)
6. Writing/co-writing legislative proposals (11%)
7. Petitions, European citizens' initiative (7%)

D. **Obstacles to transnationalization**

1. Limited know-how (15%)
2. Limited staff (11%)
3. Limited funds (7%)

E. **The European dimension**

1. Useful (70%)
2. Necessary for funding opportunities (48%)
3. Barrier against the rise of nationalist/right politics in Europe (19%)
4. Closer to younger generations in CSOs (19%)
5. Complicit in securitization of borders (19%)
6. Source of egalitarian push (15%)

## Notes

1      The interviews and other documents were transcribed and coded according to the main themes related to how CSOs depict the political environment in which they operate and the way in which they describe how their organizations are engaged in the transnational realm (for a fuller account of the coding process see the methodological Appendix A). The key objective of this discursive analysis was to identify recurring issues, themes and narratives concerning the transnational dimension of their advocacy activities. We adopted an inductive approach in coding and interpreting the texts. This methodological approach allowed us to identify emerging patterns and themes related to the practice of transnational engagement that CSOs displayed in the Italian context. Methodologically, the article relies on a discourse analysis of textual material paying particular attention to the depiction and framing of experiences of conducting political advocacy and pursuing transnational opportunities (Jørgensen and Phillips 2002).

2      All quotations from the interviews were translated from Italian by the authors.

3      Their so-called "Yellow–Green" government passed the 32/18 governmental legislation "Decree-Law on Immigration and Security" in December 2018 and the "Decree-Law on Immigration and Security n°53" of 14th June 2019, which transformed solidarity into a crime punishable by law with severe juridical consequences (Pusterla 2021, p. 79).

4      A centre-right coalition formed at its core by the Partito delle libertà (Freedoms party) and Lega Nord (the predecessor of Matteo Salvini's Lega).

5      Our translation from Italian.

6      Press release of the campaign by one of the participants: https://www.asgi.it/ingresso-soggiorno/ero-straniero-days-firme-decine-citta-cambiare-la-legge-sullimmigrazione/ (accessed on 3 June 2023).

7      The initiative saw the participation of dozens of civil society organizations of different sizes and structures, including Legambiente, The Federation of Italian Evangelical Churches, Oxfam, ActionAid, A Buon Diritto, Baobab Experience, and the Coalition for Civil Liberties and Rights (CILD).

8      More specifically, the ECI's target was the EU's Facilitators' Package and the Trafficking Directives. Its first demand was that the European Commission amend article 1(2) of the EU facilitation directive (2002/90/EC) to prevent Member States from imposing sanctions on individuals or NGOs that provide humanitarian assistance.

9      The percentages reflect how many interviews the particular themes appeared in. The categories are not mutually exclusive.

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
