# Peer review of "Europeanization as Pragmatic Politics: Italy’s Civil Society Actors Operating in the Face of Right-Wing Populism"

_socsci, doi:10.3390/socsci13040205_

Round 1

Reviewer 1 Report

Comments and Suggestions for Authors

The paper is an interesting contribution to providing deeper insights into how CSO understand the political environment in which they are acting, but also the possibilities provided to them by a European level of governance and transnational interaction. However some questions remain:

1.) The author(s) present first an overview into the newer literature on transnational activities of NGOs and on the "Europeanization" of CSO's. While this embedding of the empiricial findings in the academic literature is well written, it does not really contribute to the establishment of a clear analytical framework. It is not clearly stated, what the actual aim of the article is: is it about providing (at least anecdotal) evidence from Italian CSOs that the findings on Europeanization in the literature can be traced in the behaviour of Italian NGOs? Or is the focus on horizontal and vertical transnationalism (this is first mentioned in the empirical part, but not outlined in the theoretic part)? To make the  main thrust of the article clearer from the beginning it woul be good to provide some hypotheses or at least expectations: why (for which goals) do civil society activists seek transnational contacts? Is this different for large/small, well funded/ poorer NGOs? Under which conditions do they engage in what kind of transnationalism? Under 2.2 different characteristics of the CSOs under investigation are mentioned, but do these characteristics make a difference in engagement? Why is this broad selection of different NGOs important, if not for explaining differences in "transnationalisation"? It is stated that the purposes behind this selection was to explore how and to what extent these organizations with different sizes.... joined forces and deployed common strategies, but in the following it is all about vertical and horizontal strategies, but the differences between NGOs is not explored. 

2. It remains difficult to assess whether the statements provided in the interviews (as quoted in the text) are shared by all or most of the respondents or whether there are differences between different categories of NGOs. As a reader it would also be interesting to know more about the questions asked, not necessarily by re-stating every question, but by providing an overview of the different topics addressed during interviews, whether e.g. the actors were asked to describe their linkages with other CSOs, whether they have some institutionalized formats of cooperation and what kind etc.

3. It would be good to receive a bit more of a background to the Italian situation. The Code of Conduct is mentioned, but what is the actual content? What are the mechanisms used (by populist right-wing parties in particular) that increasingly constrains action at the national level? The reader gets only little information about the current European (legal) setting (which is particular important when the CJEU comes in), not about the specific Italian situation (is it mainly about not implementing EU rules and standards, is it about "media propaganda" to discredit human rights NGOs)? Which mechanisms of "criminalisation" are used? And is it really only "right wing" parties and populists on the national level? In the conclusion ait is stated that "Europeanization highlights the closing of political opportunities at the national level....": were there questions asked in the interviews how the situation was "before" the sucess of nationalist parties? Since when in particular are changes discernable? 

4. How well are CSOs informed about EU decision making and the role of the different institutions? Are there differences between bigger and smaller initiatives? Is there interaction only with Italian MEPs or are attempts made to interact with different MEPs of relevant parliamentary committees (LIBE or DROI)? Is the Commission generally blamed (as in the quote that "the Commission tore it all down")? The European Commission has probably little chances to act more "normatively", as the general political atmosphere in Europe (including more centrist or conservative governments) is anti-immigration ( as is also stated in the article) and while it may be possible to get the EP on board, it would be difficult for the Council. 

Reviewer 2 Report

Comments and Suggestions for Authors

Dear Authors

 the paper is interesting and it contributes important insights into the field of bottom-up civil society movements in the EU context. My only concern is related to the representation of the data. The interviewing is a qualitative, rather inductive methodological approach, however, the analysis of the conversation is shown only in terms of direct quotes, which fit predetermined thematic sections. It would be interesting if the interviews would be presented in terms of qualitative coding proces and how quotes relate to the topics of the article would come at the end of the analysis.

Round 2

Reviewer 1 Report

Comments and Suggestions for Authors

Thank you for the updated version and the cover letter. I definitely think that the horizontal coordination between the organistions and processes of learning are your most valuable argument. If anything, this could be stressed more. But otherwise you addressed the issues I mentioned. 

Author Response

Thank you for your time and effort.